# Quantitative Evaluation of Dental Students’ Perceptions of the Roleplay-Video Teaching Modality in Clinical Courses of Dentistry: A Pilot Study

**DOI:** 10.3390/healthcare11050735

**Published:** 2023-03-02

**Authors:** Kiran Kumar Ganji, Anil Kumar Nagarajappa, Mohammed G Sghaireen, Kumar Chandan Srivastava, Mohammad Khursheed Alam, Shadi Nashwan, Ahmad Al-Qerem, Yousef Khader

**Affiliations:** 1Department of Preventive Dentistry, College of Dentistry, Jouf University, Sakaka 72388, Saudi Arabia; 2Department of Oral & Maxillofacial Surgery & Diagnostic Sciences, College of Dentistry, Jouf University, Sakaka 72388, Saudi Arabia; 3Department of Prosthetic Dentistry, Jouf University, Sakaka 72388, Saudi Arabia; 4Department of Computer Science, College of Computer and Information Sciences, Jouf University, Sakaka 72388, Saudi Arabia; 5Department of Computer Science, Faculty of Information Technology, Zarqa University, Zarqa 13110, Jordan; 6Department of Public Health, Jordan University of Science & Technology, Irbid 22110, Jordan

**Keywords:** dental, perception, role play, teaching strategies

## Abstract

In the modern era of dentistry, role modeling/roleplaying is one of the most prevalent and recommended methods of dental education. Working on video production projects and using student-centred learning also help students create feelings of ownership and self-esteem. This study aimed to compare students’ perceptions of roleplay videos among genders, different disciplines of dentistry, and different levels of dental students. This study included 180 third- and fourth-year dental students registered in courses such as ‘Introduction to Dental Practice’ and ‘Surgical management of oral and maxillofacial diseases’, respectively, at the College of Dentistry at Jouf University. Four groups of recruited participants were pre-tested using a questionnaire about their clinical and communication skills. The students were tested again using the same questionnaire at the end of the workshop to evaluate improvements in their skills. The students were then assigned to create roleplay videos with respect to demonstrated skills related to all three disciplines (Periodontics, Oral Surgery, and Oral Radiology) in a week’s time. Students’ perceptions of the roleplay video assignments were collected through a questionnaire survey. The Kruskal–Wallis test was used to compare responses for each section of the questionnaire (*p* < 0.05). Improvements in problem-solving and project management skills during video production were reported by 90% of the participants. No significant difference (*p* > 0.05) in the mean scores of the responses was found with respect to the type of discipline involved in the process. There was a significant difference in the mean scores of the responses between male and female students (*p* < 0.05). The fourth year participants demonstrated increased mean scores and significantly higher (*p* < 0.05) mean scores than third-year participants. Students’ perceptions of roleplay videos differed by gender and the level of the students, but not by the type of discipline.

## 1. Introduction

Skills are the capacity to apply information quickly and effectively [1]. To be skilled in one’s craft, one must have the ability to discern. Soft skills (SS) are beneficial in both personal and professional situations. Soft skills aid in the organisation, planning, and management of changes in growing dental practise [2]. Personal values and interpersonal skills that define a person’s ability to integrate within a certain framework, such as a project team or firm, are known as soft or social skills. Individual soft skill characteristics such as communication skills [3,4,5,6], critical thinking [7], teamwork [8], leadership [9], professionalism [10], life-long learning [11], and entrepreneurship have been emphasised in dental training programs and in the literature [12]. Communication, critical thinking and problem-solving, teamwork, lifelong learning and information management, entrepreneurship, professional ethics and morals, and leadership are among the different soft skills identified by extensive research and expert consultation [13]. Perreault described soft skills as an individual’s personal traits, attributes, or level of commitment that distinguishes him from others with similar talent and expertise [14]. The implementation of soft skills by active learning tools helps not only students, by encouraging them to practise abilities and ask questions, but also professors, by enabling them to ascertain students‘ comprehension and remediate critical points in near ‘real time’ [15]. In dental education, role modeling/roleplaying is one of the most prevalent and recommended means of dental education [16]. Students’ prior role-playing experience may have an impact on how they approach this strategy [16]. In medical education, role-play is extensively utilised as an educational tool for learning about communication because it allows for observation, rehearsal, and debate, as well as realistic roles and alignment of roles with other aspects of the curriculum.

Yardley-Matwiejczuk defined it as activities in which individuals engage in ‘as if’ scenarios through simulated actions and conditions [17]. Role-playing exercises assist children in becoming acquainted with ’real-world’ events while fostering good attitudes and sentiments [18]. It gives a secure space for students to express their own and sometimes unpopular views and beliefs, and the majority of students love these activities and become more motivated learners as a result. Khalifa et al. [19] reported that student-generated videos were successful in teaching soft skills. Working on video production projects and using student-centred learning also helps students create feelings of ownership and self-esteem [20]. Omar et al. [21] studied undergraduate dentistry students’ reactions to the use of student-generated videos to teach professionalism in Malaysia. According to the authors, the majority of students thought that the movies improved their collaboration and communication skills as well as their comprehension of the dentist’s role in providing dental care.

However, there is a lack of data on students’ responses to the development of soft skills using roleplay in integrated dentistry courses. Therefore, this study aimed to evaluate students’ perceptions of roleplay videos in integrated dental courses conducted over different levels of the Bachelor of Dental and Oral Surgery program. The study hypothesised that students’ perception of roleplay videos does not differ among genders, different disciplines of dentistry, and different levels of dental students.

## 2. Materials and Methods

### 2.1. Study Design and Setting

This cross-sectional study was conducted at the College of Dentistry affiliated with Jouf University, Sakaka, Kingdom of Saudi Arabia. This study was approved by the Local Committee for Bio-ethics (reference number 14-15-9/40). A trained researcher recruited the participants and provided information about the objectives of the study. The Bachelor of Oral and Dental Surgery (BDS) degree at Jouf University is a 5-year program in which the first and second years are preclinical and the third to fifth years are clinical.

### 2.2. Participants

This study was carried out among the third- and fourth-year undergraduate dental students registered for courses such as ‘Introduction to Dental Practice’ and ‘Surgical Management of Oral and Maxillofacial Diseases’ under the integrated Bachelor of Oral and Dental Surgery program. These courses were introductory undergraduate courses of the Bachelor of Dental and Oral Surgery program at their respective levels. Both courses focused on the basic foundations of knowledge, intellectual skills, and practical skills related to various disciplines. The sample size was estimated using the G power computing tool with a margin of error of 0.05 and a critical value of 1.96. Therefore, the sample size was set at 180, assuming a response rate of 60%. This study included data from two cohorts of the study population who were enrolled for the academic years 2020–2021 and 2021–2022, and the students were informed that participation was voluntary and anonymous. A Google form was used to distribute the questionnaires to 180 undergraduate students. Cover letters explaining the study design, consent form, and importance of the study were provided to reduce nonresponse bias.

### 2.3. Procedure

In this study, students were assigned to create roleplay videos in both courses [21]. Miller’s pattern was employed with an emphasis on clinical competencies for the creation of roleplay video assignments: (1) knowledge, (2) observation, (3) simulation, and (4) experience (KOSE) [22]. The ‘knowledge’ required for the assignment creation was imparted through didactic lectures, lab sessions, and workshops. A few roleplay videos were shown to the students to familiarise them with patient communication skills as well as ethical and professional issues of the dental team. This equipped them with the convenience of learning through ‘observation’. Roleplay videos enabled learning through ‘simulation’. Learning ‘experience’ was gathered during the creation of roleplay videos as a unit in the course environment. Following the brief orientation, students were assigned to create roleplay videos with their clinical skills involving models and patients in both courses: Introduction to Dental Practice and Surgical Management of Oral and Maxillofacial Diseases. Group dynamics were followed to create a total of four groups (two groups in the third year and two groups in the fourth year). Four video assignments were given to each of these groups of students enrolled in the Introduction to Dental Practice course.

Before the students were given the video assignments, they were pre-tested using a questionnaire about their clinical skills and communication skills in the classrooms by the subject experts. Workshops were conducted for participating male and female students at the college premises. Later, they were briefed about the entire procedure through a workshop model with a demonstration of the skilled procedure that was accomplished at the student clinics by multiple faculties in groups. The students were tested again using the same questionnaire at the end of the workshop to evaluate improvements in their skills. The students were assigned to create roleplay videos with respect to all three disciplines (Periodontics, Oral Surgery, and Oral Radiology) about the demonstrated skill procedure in a week’s time. Similar procedures were practised in all three courses for the creation of roleplay videos. These video assignments were aimed at instilling basic lifelong clinical and communication skills in various disciplines. The students were given common clinical scenarios and a stipulated time for the completion of the video assignments. The students shoot/created video assignments at the College of Dentistry’s student clinics. Furthermore, the students worked collectively and processed the videos themselves to ensure originality, quality, and timeline. Following the submission of the roleplay videos by the students, the same were graded by experienced faculties of respective disciplines for the content, and feedback was given to each group using predetermined rubrics.

### 2.4. Study Instrument

The students’ responses on the creation of the roleplay video assignments were collected through a questionnaire survey shared via Google Forms [21]. The study instrument consisted of six sections: (1) effectiveness of instructional videos, (2) general satisfaction with instructional videos, (3) open-ended questions, and (4) how satisfied were you with the introduction of educational videos in the course? A five-point Likert-Scale was used for the first two items (strongly agree, agree, neutral, disagree, strongly disagree). Similarly, a five-point Likert scale was used to score the items (poor to excellent). The questionnaire distributed to the participants was developed, discussed, and reviewed with co-investigators for its relevance to the course and regional cultural adaptation, as described by Artino et al. [23]. The questionnaire was pre-tested for its validity and reliability by expert evaluation before distribution to the study participants.

### 2.5. Study Variables

Participants’ responses to the questionnaire were used as the outcome data. Gender, year of study, and subject discipline were selected as associated factors in the present study. Data on the sex, year of study, and subject discipline of participants were collected using a self-report survey.

### 2.6. Statistical Analysis

For each study variable, the mean, standard deviation, frequency, and percentage were calculated. The mean score was calculated by averaging the options agree (3), neutral (2), and disagree (1) for each item. The Brunner Munzel test was administered to compare male and female responses and students with different courses. We used the Kruskal–Wallis test to compare the responses for each section of the questionnaire. *p*-values of 0.05 were considered statistically significant for statistical analysis using SPSS version 22 (IBM, Armonk, NY, USA).

## 3. Results

In total, the survey involved 180 students over two cohorts (third- and fourth-years) of a Bachelor of Oral and Dental Surgery program. In general, most students perceived the advantages of role-playing as an effective teaching method for improving their dentistry skills. As part of the ‘student preparation for roleplay production’ project, most students reported that preparing the scripts helped them grasp the written material in the lesson (90%) and gain confidence in demonstrating the skill (89%). Ninety percent of the participants reported an improvement in their problem-solving and project-management skills during video production (Figure 1, Figure 2 and Figure 3).

A comparison of participant feedback responses across the three disciplines revealed no significant difference (*p* > 0.05) in responses to all items, indicating no differences in the rating response between disciplines. Roleplay, as a teaching tool for video production and soft skill development, was unaffected by the type of discipline, as all three disciplines were rated highly by the participants (Table 1).

Table 2 compares the responses of participants by gender. The mean scores for all questionnaire items were higher for female students than for male students, indicating a more positive outlook. There was a significant difference in the mean scores for these seven items between the male and female students (*p* < 0.05).

Table 3 presents comparisons of the participants’ responses regarding the course level (third- and fourth-year). The participants at the fourth-year level demonstrated increased mean scores and significantly higher (*p* < 0.05) mean scores than participants at the third-year level for all seven items (*p* < 0.05).

## 4. Discussion

Simulations such as roleplaying are used to teach knowledge, attitudes, and skills in a variety of fields. Involving students in the learning process makes this teaching method enjoyable and active [24,25]. Roleplay has also been found to improve students’ critical thinking and attention span compared to lectures, demonstrations, tutorials, and field studies [25]. Medical educators utilise roleplays to teach students communication skills. Through roleplaying, students become familiar with roles/characters, discuss them with others, and practise them, aligning the topics with the course learning outcomes. Introducing students to ’real world’ scenarios through role play helps them gain positive attitudes and feelings. Furthermore, it provides students with an opportunity to express themselves and learn unpopular attitudes and opinions, and the majority of students find these activities stimulating and inspiring. The guidelines mentioned in Mogra et al. are based on Pendleton’s rules, which are considered ways to ensure that feedback is constructive rather than destructive [26]. In addition to improving cognitive abilities and psychomotor skills, roleplay assignments also have positive effects on the learning outcomes of dental programs. Research has shown that student-generated videos have positive effects on learning [21]. The present study compared students’ perceptions of roleplay videos implemented in three different courses across two levels of BDS students. Throughout the study period, the team of instructors, methods of delivery for classroom instruction, themes of roleplay assignments, and measuring instruments remained the same. The conditions needed for appropriate comparisons among the three disciplines in the study, namely periodontology, Oral Surgery, and Oral Radiology. Al-Khalifa et al. showed that video assignments should be a part of professional courses in dentistry to improve students’ communication skills with their colleagues and patients [27]. However, no studies have evaluated the efficacy of roleplay teaching methodologies in the clinical course of dentistry. Therefore, the current study was the first to evaluate participants’ perceptions of roleplay teaching strategies for individual disciplines in an integrated dental curriculum course. The findings of the study showed that most students recognised the importance of roleplay videos by the end of the study. Students’ perceptions of roleplay video teaching modes were found to vary by gender and student level, with female students expressing highly favourable views of this teaching modality. The students’ perceptions of roleplay videos in dentistry did not differ across disciplines. The findings of the study showed that participants’ perception of roleplay was very high with respect to Periodontics, Oral Surgery, and Oral Radiology. The sole purpose of choosing these three disciplines was based on the concept that they are horizontally integrated within an introductory course for third-year and fourth-year BDS programs at the College of Dentistry, Jouf University. Upon evaluation, there was no significant difference in the participants’ questionnaire response scores for these three disciplines, indicating that the roleplay teaching methodology was beneficial in all respects. Nestel and Tierney reported that students learned communication skills through roleplay [16]. Manzoor et al. found that 88.9% of medical students were satisfied with the communication skills acquired through roleplaying. These findings are in agreement with a recent report by Manzoor et al. [25]. As a result, students who were exposed to a new course viewed roleplaying video assignments as a powerful learning tool because roleplaying can enrich learning experiences through emotional engagement, movements, and variations.

The current study found that female students preferred roleplay as a teaching method for allied clinical courses, which was confirmed by a higher rating than by male participants. Such gender differences are due to different preferences for learning styles among dental students. Numerous studies have indicated that female students learn better through kinaesthetic approaches [28,29,30,31]. Similarly, a greater proportion of students with high academic performance reflected through gross point average (GPA) scores demonstrated a preference for a kinaesthetic learning style than those with low academic performance [32]. Surprisingly, in our study, females with kinaesthetic learning style preferences also had higher GPA scores. Kinaesthetic learners learn through practise, experimentation, and experience and prefer learning via roleplays, field trips, and case studies [33].

With respect to participants’ perception level (third- and fourth-year) towards roleplay, it was estimated that fourth-year students had a more positive view of the importance of roleplay in understanding textbook information more deeply. This is in contrast to a study conducted by Al-Khalifa et al. [19], which demonstrated that significantly higher proportions of third-year students than fourth-year students thought that roleplay improved the clarity of communication, provoked critical thinking, increased their attention span, and was a feasible method of learning. Such variation in perception is related to the type of course in which the roleplay was implemented. As noted in the current study, the fourth-year students understand the importance of roleplay in clinical courses, and they also realised the key aspects of roleplay through learning experiences after getting involved in them. In the current study, the students’ ability to apply their content knowledge to new clinical situations was improved by modelling application skills, as shown in the videos prepared by the students. These findings are in accordance with the reports of Miller et al. and Kavadella et al. [34,35]. The proceedings of the study could illuminate the need for advanced care planning to enhance the learning experience of dental students in an integrated dental curriculum, as suggested by Blomberg et al. [36]. Role-play is indeed a highly effective method of teaching communication skills, but given time limitations on the part of the faculty, it sometimes becomes difficult to practise many practical scenarios. We targeted and addressed a major lapse of such situations in our undergraduate training and strongly agree with the fact that it is the need of the hour to train dental undergraduates to encourage imagination, creativity, and a sense of active learning strategies among peers. We also believe that good communication skills with problem-solving skills employed through role play need to be taught to dental undergraduates to inculcate in them greater clinical competence and that these skills can never be taught through didactic lectures. Our belief is that video-assisted learning would overcome these limitations and would enable students to be exposed to a variety of scenarios via roleplay with limited faculty and fewer time constraints. The current study encountered some limitations while teaching through roleplay methods, such as the topic of roleplay varying for different levels of participants and time management. With roleplay activities, students deviate towards directed self-learning instead of self-directed learning, which would lead the students to become involved in information that is scientifically not evident. Hence, peer involvement is mandatory for the successful completion of roleplay activities.

## 5. Conclusions

Clinical courses in dentistry should include video assignments as part of the curriculum to improve students’ communication skills and professional behaviours with their colleagues and patients. With the aid of technology, dental students can develop problem-solving skills whenever they roleplay together as a group.

## Figures and Tables

**Figure 1 healthcare-11-00735-f001:**
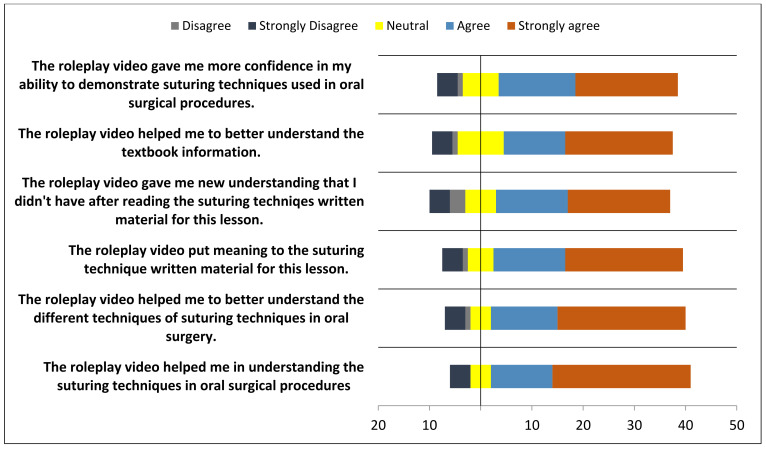
Responses of the participants towards roleplay soft skills in Oral Surgery discipline.

**Figure 2 healthcare-11-00735-f002:**
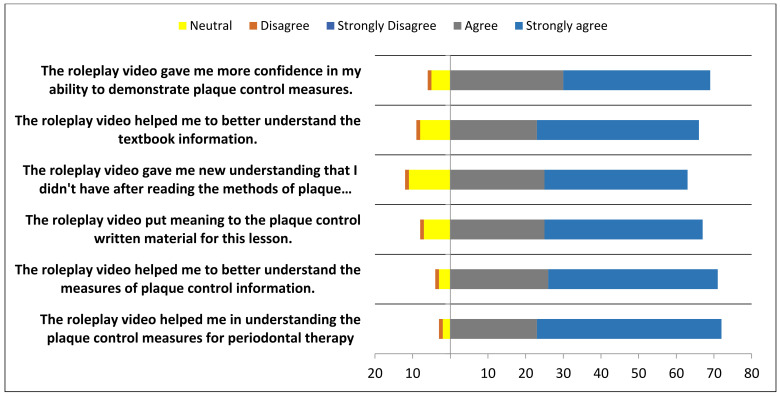
Responses of the participants towards roleplay soft skills in Periodontics discipline.

**Figure 3 healthcare-11-00735-f003:**
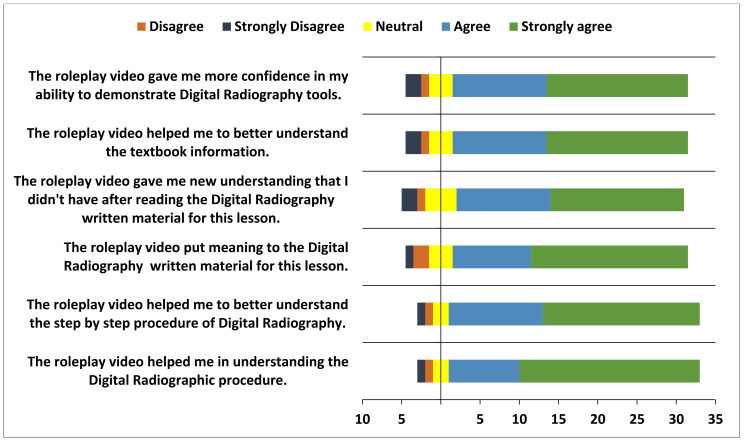
Responses of the participants towards roleplay soft skills in Radiology discipline.

**Table 1 healthcare-11-00735-t001:** Comparison of feedback responses for roleplay videos implemented in various disciplines of the BDS course.

Questionnaire		Periodontics Discipline	Oral Surgery Discipline	Oral Radiology Discipline	*p* Value
Sections	Sub-Section	Mean ± SD	Mean ±SD	Mean ± SD	
Preparation of students for video production	a. This roleplay video gave me new understanding that I didn’t have after reading the XYZ written material for this lesson.	2.75 ± 0.60	2.73 ± 0.48	2.68 ± 0.63	0.094
	b. This roleplay video helped me to better understand the textbook information.	2.86 ± 0.66	2.76 ± 0.51	2.84 ± 0.66	0.084
	c. This roleplay gave me more confidence in my ability to XYZ.	2.83 ± 0.74	2.76 ± 0.72	2.79 ± 0.71	0.076
Skill development achieved from roleplay videos	a. My language and communication skills improved as a result of participation in video production	2.84 ± 0.48	2.81 ± 0.51	2.79 ± 0.43	0.083
	b. My problem-solving skills were improved by participating in video generation	2.76 ± 0.60	2.84 ± 0.63	2.82 ± 0.66	0.072
Overall learning experience from roleplay videos	a. Overall I was satisfied with these roleplay videos.	2.86 ± 0.72	2.83 ± 0.66	2.89 ± 0.69	0.085
	b. Overall I feel I was able to learn the information from this roleplay videos as well as I would have in a face-to-face class presentation.	2.81 ± 0.77	2.78 ± 0.74	2.80 ± 0.72	0.094

**Table 2 healthcare-11-00735-t002:** Comparison of feedback responses for roleplay videos in relation to gender of the participants.

Questionnaire		Male	Female	*p* Value
Sections	Sub-Sections	Mean ± SD	Mean ± SD	
Preparation of students for video production	a. This roleplay video gave me new understanding that I didn’t have after reading the XYZ written material for this lesson.	2.79 ± 0.40	2.95 ± 0.58	0.038 *
	b. This roleplay video helped me to better understand the textbook information.	2.76 ± 0.46	2.85 ± 0.71	0.005 *
	c. This roleplay gave me more confidence in my ability to XYZ.	2.77 ± 0.61	2.94 ± 0.63	0.021 *
Skill development achieved from roleplay videos	a. My language and communication skills improved as a result of participation in video production	2.83 ± 0.63	2.91 ± 0.71	0.015 *
	b. My problem-solving skills were improved by participating in video generation	2.64 ± 0.83	2.85 ± 0.74	0.005 *
Overall learning experience from roleplay videos	a. Overall I was satisfied with these roleplay videos.	2.85 ± 0.68	2.90 ± 0.74	0.003 *
	b. Overall I feel I was able to learn the information from this roleplay videos as well as I would have in a face-to-face class presentation.	2.84 ± 0.69	2.92 ± 0.59	0.000 *

* statistically significant (*p* < 0.05).

**Table 3 healthcare-11-00735-t003:** Comparison of feedback responses for roleplay videos in relation to third-year and fourth-year levels of BDS students.

Questionnaire		3rd-Year	4th-Year	*p* Value
Sections	Sub-Sections	Mean ± SD	Mean ± SD	
Preparation of students for video production	a. This roleplay video gave me new understanding that I didn’t have after reading the XYZ written material for this lesson.	2.72 ± 0.50	2.89 ± 0.68	0.016 *
	b. This roleplay video helped me to better understand the textbook information.	2.83 ± 0.86	2.95 ± 0.71	0.019 *
	c. This roleplay gave me more confidence in my ability to XYZ.	2.71 ± 0.44	2.86 ± 0.56	0.008 *
Skill development achieved from roleplay videos	a. My language and communication skills improved as a result of participation in video production	2.82 ± 0.61	2.90 ± 0.39	0.015 *
	b. My problem-solving skills were improved by participating in video generation	2.66 ± 0.70	2.84 ± 0.59	0.005 *
Overall learning experience from roleplay videos	a. Overall I was satisfied with these roleplay videos.	2.81 ± 0.52	2.93 ± 0.80	0.002 *
	b. Overall I feel I was able to learn the information from this roleplay videos as well as I would have in a face-to-face class presentation.	2.79 ± 0.69	2.89 ± 0.59	0.000 *

* statistically significant (*p* < 0.05).

## Data Availability

Data will be made available upon request to the corresponding author.

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
