# Peer review of "Quantitative Evaluation of Dental Students’ Perceptions of the Roleplay-Video Teaching Modality in Clinical Courses of Dentistry: A Pilot Study"

_healthcare, 2023, doi:10.3390/healthcare11050735_

Round 1

Reviewer 1 Report

Dear Authors

Thank you very much for the submission of the manuscript.

In general, the study fits into the scope of the journal and represents scientific soundness. However, some minor issues need to be addressed.

(1) Some info is missing in the Abstract: Please add info related to the number of students included, the statistical methods used, and provide exact p-values. 

(2) In scientific publications it is recommended not to use personal pronouns, instead use "passive voice". For example, page 9, line 251: "Our Study...". This should be checked throughout the manuscript and modified accordingly.

Author Response

Dear Reviewer,

We thank for your valuable comments. All the comments raised by the respected reviewer are being addressed using point to point clarification table attached herewith. Kindly refer the same.

Regards

Reviewer 2 Report

I have reviewed the manuscript “Quantitative evaluation of dental students’ perception to roleplay videos teaching modality in clinical courses of dentistrycurriculum: A Pilot study”. The study aimed to compare the student perception of roleplay videos in dental teaching.

I recommend its publication only after addressing the following issues and suggestions:

1.    INTRODUCTION

-      This section should provide more information and include more relevant references.

-      In the text, reference numbers should be placed in square brackets [ ], and placed before the punctuation, not after the punctuation (see lines 44, 47, 52, 60, 63, 75, 79)

-      Line 72: “Khalifa et al” >  please, change with: “Khalifa et al. [19] “

-      Line 75: “Omar et al.” >  please, change with: “Omar et al. [21]”

2.     MATERIALS AND METHODS 

-      Please provide the Project identification Code (Protocol number) and date of approval (see line 90). As a minimum, a statement including the project identification code, date of approval, and name of the ethics committee or institutional review board must be stated in Section ‘Institutional Review Board Statement’ of the article.

-      Please include in the “Participants”  (or “Selection of participants”) sub-section the number of students involved in the study (from Results section- see line 145) 

-      The methodology presented by the authors is not adequately describedand the research design should be improved.

-      Please, reconsider this section in a more comprehensible manner.

3.    RESULTS

-       Questionnaire’s design should be improved. The questions’ wording should not influence students’ opinions and should be objective and appropriate; avoid including leading questions in a questionnaire, so the obtained results won’t be predictable and will add value to the study

4.    DISCUSSION

-       This section should include more relevant  recent references.

5.     INSTITUTIONAL REVIEW BOARD STATEMENT:

-      Please provide the Project identification Code (Protocol number) and date of approval. As a minimum, a statement including the project identification code, date of approval, and name of the ethics committee or institutional review board must be stated in Section ‘Institutional Review Board Statement’ of the article.

6.    REFERENCES

References need to be revised, for instance:

-      Duplicated references: Reference 19. And 26 Al-Khalifa, K.S.; Gaffar, B.O. Dental students’ perception about using videos in teaching professionalism: A 307 Saudi Arabian experience. J. Dent. Educ. 2021, 85, 197-207.

-      More recent references are needed

7.    Extensive editing of English language and style is required.

8.    Authors should explain the novelty or relevance of their study.

9.     I recommend the publication of this manuscript after major revisions.

Author Response

(The authors gave the same response as above.)

Reviewer 3 Report

Dear authors,

This article is interesting, however, some comments should be addressed:

Keywords should be arrange according to mesh terms and alphabetical order.

Lacks of hypothesis

results section should be done with level of significane. Letters should be added to tables.

Discussion section needs to be improved by adding more limitations. The use of "our" should be avoided in the manuscript.

The conclusion section is a repetition of the results, so please reformulate

Author Response

(The authors gave the same response as above.)

Round 2

Reviewer 2 Report

I have read the revised version of the manuscript: “Quantitative evaluation of dental students’ perceptions of the roleplay-video teaching modality in clinical courses of dentistry: A pilot study”

I do appreciate authors’ effort to improve the quality of the manuscript. The manuscript has been sufficiently improved to warrant publication in Healthcare. However, this revised version of the manuscript contains minor issues that necessitate clarification, such as:

1.    It is recommended that the title be written using capital letters in accordance with accepted scientific norms. Please ensure that the title is revised accordingly. Example: “Quantitative Evaluation of Dental Students’ Perceptions of the Roleplay-video Teaching Modality in Clinical Courses of Dentistry: A Pilot Study” 

2.    In accordance with scientific convention and the journal’s “Instructions for Authors”, reference numbers should be enclosed within square brackets [ ] and positioned prior to any punctuation marks. It is imperative that this formatting guideline be adhered to throughout the entirety of the manuscript. Kindly effectuate the necessary modifications accordingly. Example: 

“Skills are the capacity to apply information quickly and effectively.[1]” 

Please effectuate the following modification: 

“Skills are the capacity to apply information quickly and effectively [1].”

3.    Line 166: “F or” > Please effectuate the following modification > “For”

4.    Please provide clarification on the time frame of the study. Additionally, could you please clarify the following phrases / elaborate on the intended meaning of the following phrases:

Line 89-91: “This cross-sectional study was conducted at the College of Dentistry affiliated with Jouf University, Sakaka, Kingdom of Saudi Arabia, between September and November 2021.”

Line: 106-107: “This study was conducted during the academic years 2020–21 and 2021–22…”

5.    Please verify the numerical ordering of the bibliographic references subsequent to the rectification of the duplicate entry. 

I recommend the publication of this manuscript after minor revisions.

Congratulations to the authors and best of luck!

Author Response

Dear Reviewer,

Thank you very for your feedback and encouragement. All the recommendations proposed are hereby incorporated in the revised version of the manuscript. A point to point clarification file is also provided herewith.

Regards

Corresponding author.

Reviewer 3 Report

Thank you for the revised manuscript. All references should be done before punctuations please correct.

Author Response

Dear Reviewer,

Thank you very for your feedback and encouragement. The recommendation proposed  hereby  are incorporated in the revised version of the manuscript. Reference formatting is being rectified to be as per the author guidelines of the journal.

Regards

Corresponding author.